adolescents; Kenya; depression; anxiety; schizophrenia

**Corresponding author:**
Pamela Wadende;
Email: pamela.wadende@gmail.com

# Mental health literacy: Perspectives from Northern Kenya Turkana adolescents

Pamela Wadende[1] and Tholene Sodi[2]

[1]School of Education and Human Resource Development, Kisii University, Kisii, Kenya and [2]Department of Psychology, University of Limpopo South Africa, Sovenga, South Africa

## Abstract

Mental illness accounts for high levels of morbidity, mortality and poor quality of life among young people. Depression, anxiety, conduct issues and hyperactive disorder account for 13% of the global burden of disease that affects one in seven adolescents. In Kenya, not much is documented about the mental health of non-school-going adolescents, and yet they make up about 1.8 million of the country's population.

An ethnographic study by Focus Group Discussions with 32 rural northern Kenya Turkana-based adolescents divided into school and nonschool groups was conducted. We read out vignettes in which the main character exhibited symptoms of depression, schizophrenia or anxiety and explored their knowledge of causes and management options for the same, and then analyzed the data thematically.

Participants described the conditions without referring to the local names we had collected earlier: depression (*Akiyalolong*), schizophrenia (*waarit/Ngikerep*) and anxiety (*Ngatameta naaronok*). They assigned curses, guilt, hunger pangs, and evil spells as causes, and believed friends and age-mates, parents, teachers, and the local chief, among others, could help, but rarely medical intervention. Interventions to improve the adolescent's knowledge of mental illness are a much-needed support for the health of young people.

## Impact statement

*Background to the problem*: Mental illness accounts for high levels of morbidity, mortality and poor quality of life among young people. Globally, one in seven youth between 10 and 19 years are mentally ill, making up 13% of the global burden of disease; depression, anxiety, conduct disorders and attention-deficit/hyperactivity disorder lead here. Unaddressed mental illness progresses into adulthood disrupting victims' lives. In Kenya, most studies on various aspects of child mental health have focused on school-going adolescents, such as Ndetei et al. (2009, 2010, 2022) and Kamanderi et al. (2019). Not enough is documented of both school- and non-school-going adolescents and children in Kenya, even while it is estimated that the number of non-school-going children is about 1.8 million. To seek help for mental illness, one has to recognize and understand its causes.

*Actions to address the problem*: We conducted an ethnographic study to explore rural-based Kenyan mental health literacy for depression, anxiety disorders and schizophrenia. We decided on the first two as they were already identified as leading mental illnesses, and schizophrenia which was not prominently identified, but is one most stigmatized in low- and middle-income countries with limited capacity to engage mental illness in general.

*Beneficiaries of our project*: Our project with both school- and non-school-going adolescents shifts focus onto an area that has had limited research in Kenya as well as in other low- and middle-income countries. The findings of our study among the 32 participants from the arid pastoralist Turkana community are important for researchers in adolescent mental health. They can use our methods and findings to conceptualize their future projects. Interventionists interested in adolescent mental health will use our findings to develop appropriate intervention programs.

## Background

Mental illness is becoming a highly prevalent health problem that is associated with high levels of morbidity, mortality and poor quality of life among young people. Global estimates show that one in seven or 14% of youth between 10 and 19 years suffer a mental health condition, accounting for 13% of the global burden of disease (Institute of Health Metrics and Evaluation, 2019; WHO, 2021). Research has it that depression, anxiety, conduct disorders and attention-deficit/hyperactivity disorder (ADHD) are the leading causes of health-related disability in this age group (Institute of Health Metrics and Evaluation, 2019; WHO, 2021). Unaddressed, the

problems continue into adulthood, affecting one's relationships, work and overall quality of life. The recent outbreak of the COVID-19 pandemic has exacerbated this situation and has resulted in more psychological vulnerabilities for many individuals and communities globally. Studies show the need for timely interventions to address the mental health needs of young people (WHO, 2013; McGorry and Mei, 2018; Colizzi et al., 2020) to ensure a healthy adulthood for the youth.

In Kenya, most mental health studies among the youth have focused on school-going adolescents, such as Ndetei et al. (2022) on suicidal ideation and illegal drug use (Ndetei et al., 2009, 2010; Kamanderi et al., 2019). In a study with 9,742 high school, college and university-level youth aged between 15 and 25 years. Ndetei et al. (2022) found that 22.6% of the sampled youth had different types of suicidal ideation, while 20.0% had major depressive disorders. Indeed, it is noted that when the youth live in volatile conflict and deprived situations, such as in many low- and middle-income country (LMIC) contexts, they become especially vulnerable to mental ill-health (Kieling et al., 2011). Apart from the limited capacity of the health systems in LMICs such as Kenya to engage mental illness, stigma is often an impediment for the sick to seek treatment (Osborn et al., 2021). It is possible to mitigate the problem of inadequate front-line workers able to support adolescents suffering from mental illness. Venturo-Conerly et al. (2021) tested a low-cost, non-stigmatizing, accessible community-based intervention mental health program for school-going adolescents in Kenya, delivered by lay persons. This was to circumvent the problem typically felt by countries in Sub-Saharan Africa where there is limited adequately trained mental health personnel to serve populations. This intervention was shown to significantly reduce symptoms of depression and anxiety, and improve social support and academic performance relative to the control group. Studies such as the preceding one could also be conducted with non-school-going adolescents to gauge the efficacy of these interventions.

A growing body of research suggests that interventions such as mental health literacy can contribute significantly toward promoting overall mental well-being in young people as it increases the probability that individuals will seek professional mental health care for themselves and others (WHO, 2013; McGorry and Mei, 2018; Colizzi et al., 2020). As reported in some of the above sections, there is insufficient curated data to assess the burden of mental illness in Kenya. Our study aimed to contribute this data by surveying mental health literacy among school- and non-school-going (locally known as *Raia*) adolescents between the ages of 13 and 17 among the pastoralist Turkana of remote northern Kenya. The two groups of adolescents, who we disaggregated further by gender, not only co-exist in one communal environment but often in one family. We felt it was important to also survey the situation of the Kenyan out-of-school adolescents that make up most of the 1.8 million children and adolescents aged between 6 and 17 years estimated by the UNESCO Institute of Statistics report (UNESCO Institute of Statistics, 2021). The two groups of adolescents live side-by-side but in two different worlds where they have largely different experiences because of their 'calling' (school or 'Raia').

## Aim

The aim of the study was to conduct a mental health literacy survey among selected adolescents in northern Kenya.

Main questions:

1. What do adolescents know about mental illness in terms of its conceptualization, causes and management?
2. What support systems do they have for their mental health?

## Methods

### Study settings

The study was conducted in northern Kenya in the rural Turkana West constituency's Lokichogio ward. Here, families commonly keep some selected children away from school so they can continue the pastoral lifestyle of herding livestock. Boys herd animals, while girls are prepared to be married off in exchange for livestock. The children thus retained at home are locally referred to as *Raia*, which is a Swahili word meaning 'citizen'. This reference denotes their superior capacity to retain local indigenous knowledge necessary for survival in the challenging environment characterized by livestock raids from hostile neighbors, fight over grazing grounds and water, poor nutrition because of incessant drought and, most of all, high levels of poverty and inequity. According to the Kenya National Bureau of Statistics (KNBS), Turkana is the poorest of the 47 counties of Kenya, with approximately 88% of the population living below the national poverty line, in comparison with the national rate of 45% (KNBS, 2014).

Despite the challenges in Turkana land, they are one Kenyan ethnic community that hold onto their traditions despite lifestyle-changing events such as prolonged exposure to other dominant ways of life as Operation Lifeline Sudan (OLS) (Akol, 2005). OLS was an altruistic operation between 1990 and 2005 through which the UN worked directly with armed local groups and NGOs to avail humanitarian aid to people affected by armed conflict without conferring formal recognition to the armed groups. The effort based its operations in Lokichogio town and transformed it into an epicenter of multiple daily flights, influx of people from high-income countries and Kenyans from other, more developed (as far as infrastructure and education are concerned) parts of the country. The influx of strangers had some undesirable consequences such as an increase in thievery and prostitution, among other social ills. Despite this, 16 years later, much of life in this area is quite traditional as local Turkana still holds onto their culture in their forms of dressing, food, beliefs, rituals and conceptualizations of what is good and what is bad (Akol, 2005).

### Study design

This was an ethnographic study that involved an elaborate familiarity with the study site and participants before the study began. We opted for ethnography as our method of study because we were working among a mixed rural population of school- and non-school-going adolescents. Ethnography would allow us to collect their individual opinions on the topics at hand. Our study utilized Focus Group Discussions (FGDs) to collect data. Jain and Orr (2016) surveyed ethnographic perspectives on global mental health status and found that this research method yielded nuanced evidence on mental health's various manifestations across different settings and accounted for other determinants of distress such as socioeconomic status (Hofstede, 2001; Mills, 2014; Mills and Fernando, 2014; Tribe, 2014), among other findings. We sought to explore participants' understanding of mental health conditions such as depression, schizophrenia and anxiety within their culture

using this method because one of ethnography's strengths is its ability to reveal and highlight the significance of social processes that surround the study participants (Jain and Orr, 2016). Like other African communities, the Turkana uphold a collectivist culture, which is a 'we' as opposed to an 'I' individualistic way of life, and consider themselves as belonging to a whole with intertwined fates where they share and care for each other (Hofstede, 2001). Scholarship in the African continent has shown that despite the multi-ethnic nature of the continent (Sanda, 1978), there is a general similarity in life experiences and cultural traits shared by various African societies, one of which is the acknowledgement of the supremacy of the group over the individual. Indeed, Patton (2002) and Krueger and Casey (2009) note that FGDs have been used successfully to collect data in collectivist cultures mostly because it is unlikely for researchers to get the privacy they need to interview a single individual since other community members would often want to listen in. Individual interviews were thus almost impossible to schedule as also found by Patton (2002), who reports of similar experiences in Tanzania and Burkina Faso. Additionally, the Turkana community particularly takes matters of security very seriously because of its common experience of animal raiding and general armed conflict with some of its neighboring communities. It follows that they are uncomfortable with activities/procedures that require secrecy as individual interviews would.

Furthermore, the adolescents were divided into four homogenous groups to allow for free expression by the group members. These were school-going boys, school-going girls and the same division for non-school-going adolescents. The sociodemographic data collected focused on such items as age, school or non-school-going status and family background, including how many siblings they had. Individual responses were collected in FGDs to make it easier to link responses to both individual and their sociodemographic information during data analysis. During the FGDs, each person was asked to mention the pseudo names earlier assigned to them before they made any contribution to the discussion.

Earlier before data collection, our team got permission from relevant officials. These included the education and local administration officials as well as a research institution ethics approval for our proposal. Our research assistants, in addition to being literate in English and Kiswahili, the official and national languages, respectively, could speak and write the local Turkana language. Additionally, the study FGDs were held where the participants would feel most comfortable and this was at the local administration offices. The aura here was calming as shown by the ease with which locals accessed the place to water their animals from the water borehole sunk at these grounds for just this purpose. Children also comfortably made mud toys as they waited for their animals to drink water.

## Participants

A total of 32 boys and girls aged between 13 and 17 years participated in the study; making up four groups (eight school-going adolescent boys, eight school-going adolescent girls, eight non-school-going adolescent boys and eight non-school-going adolescent girls). It was necessary to have these four-participant groups of adolescents because of the realities of Turkana lifestyles where some children are identified and kept home from school to continue with the pastoralist way of life, which is majorly raising domestic animals. Each of the four-participant groups was interviewed separately in FGDs so as to maximize on information collected by minimizing any inhibitions that could come up in mixed gender, mixed social exposure (as would happen if school- and non-school-going adolescents were mixed in a discussion group).

## Measures

The study used vignettes of fictional characters to start discussions and for a long time as a stimulus that encouraged study participants to air their thoughts freely (Jenkins et al., 2010; Sampson and Johannessen, 2020). Although these fictional vignettes are powerful instruments that have been used to explore social norms and values, some researchers have raised reservations on how to pick out participants' responses that represent their own views and ideas and those that represent their understandings of social norms (O'Dell et al., 2012) and idealized responses that have no bearing to reality (Barter and Renold, 2000). It can, however, be argued that researchers utilizing any data collection instrument can never be absolutely sure that participants' responses represent their actual views. In using these vignettes, we draw on their already established advantages and especially for their ability to spur free discussion among participants in an interview. We believe that vignettes being snippets of stories are familiar to a rural community such as this one we worked with because they hold onto much of their traditional lifestyle, which include oral story-telling.

Our survey adapted items assessing each element of mental health literacy, namely: recognition and knowledge, etiology and management of mental illness previously surveyed both among Canadian young adults (Marcus and Westra, 2012) and in the Scottish National Survey of Mental Health Attitudes using questionnaires (Glendinnin, 2002). These surveys in Canada and Scotland began with a vignette of individuals (randomly assigned a context-relevant name) suffering from what are typical symptoms of depression, anxiety or schizophrenia (also randomly selected with related symptoms) according to the *Diagnostic and Statistical Manual of Mental Disorders-IV* (American Psychiatric Association, 2013) criteria. Our study gave the fictional characters local names (Akuam, Lokobe and Eragae in place of the Mary, Robert and Mary as used with the Canadian mental health literacy survey). Apart from changing the names of the main characters, we kept the story in the vignettes the same since they contained human feelings common all over the world.

FGDs seem more feasible for the literacy levels of our mixed in- and out-of-school adolescents and the collectivist nature of the local community (Hofstede, 2001). As already noted, various scholars recommend using FGDs with participants from collectivist cultures (Patton, 2002; Krueger and Casey, 2009). Additionally, our study assumed that adolescents would be more interactive when involved in discussions with their peers, which in this case were the different strata in the sample; school-going girls, non-school-going girls, school-going boys and non-school-going boys.

We developed probing questions for the scenarios, both anticipatory (with the investigating team) and during interview session requests for more information as follow-up items. Of the three common probing techniques, accommodation, encouraging and challenging questioning (Moerman, 2014; McGrath et al., 2019), accommodation and encouraging techniques of probing participant responses seemed the best for our study. Here the study affirmed what the adolescent had said and encouraged them to say more with open-ended follow-up questions when they seemed not to have exhausted their thoughts on a survey item.

### Translation process of vignettes from English to Turkana

The study followed a five-step translation process for the vignettes with our research team members (Beaton et al., 2000; Checa et al., 2018; DuBay and Watson, 2019; Hallit, 2020). Our team included native Turkana speakers who were also fluent in English and experienced in conducting translations for research and court cases, conducting their own research, working and for non-governmental organizations working in Turkana County. First, there was a forward translation of the documents with two of our team members, each making an independent translation. Second, our investigating team synthesized the two translated documents through the help of a third translator who looked for words that were different in the two presented translations but hardly got any. In the third step, the team back-translated the vignettes and conducted validity checks to ensure the synthesized translation reflected the same item content as the original English version. Here the translation and checks were done by a native speaker who works as an administrator in the judiciary and has experience translating for courts and research projects in the area. In the fourth stage, our whole translating team met and discussed the document to ascertain that it was ready for pilot testing as the last translation stage. Here the translated document and the English version were read by a senior scholar with a Ph.D. in childhood studies who is a native Turkana speaker and had done many studies among different communities in this area. Our team pilot-tested the translated measures and got ready to collect data.

All FGD sessions started with a narration of the vignettes of imaginary persons suffering from depression, schizophrenia or anxiety as a catalyst for a discussion on the same. The main characters in the vignettes took on Turkana names: The first character, a girl suffering from depression, was given the name Akuwam. The second character, a man suffering from schizophrenia, was given the name Lokobe. The last character who is a girl suffering from anxiety was named Eragae. Both the names and health conditions were randomly selected from the local context and the *Diagnostic and Statistical Manual of Mental Disorders-V* (American Psychiatric Association, 2013) criteria, respectively. The three vignettes presented the following scenarios:

1. Akuwam has been feeling really down for the last few weeks. She finds it hard to concentrate on anything, and has no energy at all. She is feeling so down that she has lost her appetite, has trouble sleeping and has been unable to go to work.
2. Over the last few months, Lokobe has become convinced that people are spying on him and can hear what he is thinking. He has lost interest in work and family activities and is spending most of the day in his room. He has been hearing voices, even though nobody is around.
3. Things bother Eragae more than they bother other people, and she worries a lot and is nervous much of the time.

### The data analysis process steps

Our research team adopted the Braun and Clarke (2006) and Braun and Clarke (2014) six-phase framework for conducting thematic analysis. The following steps were followed once the data were collected: transcription, organizing data, coding, deducing categories, identifying common themes and making interpretations, and keeping a reflective journal.

### Findings

Below are the main themes and related quotations from participants from the different categories of adolescents in this project: non-school-going boys (*Raia*), school-going boys, non-school-going girls (*Raia*) and lastly school-going girls.

### Conceptualization of depression, schizophrenia and anxiety

#### Depression 'Akiyalolong'

Although none of the 32 adolescents who took part in the FGDs pointed out that it could be depression causing Akuwam's symptoms, they, however, described what was Akuwam's situation. They thought this was extreme sadness caused by unfortunate happenings in Akuwam's life. The nearest they came to acknowledging that it was a mental illness was when one 'Raia' boy alluded to mental illness, without mentioning the particular illness.

*According to me, Akuwan has a mental illness, and that is what disorients her a lot,* Raia boy 1 (15).

Others attributed it to exclusion from society, fear of imagined harm, grief, family conflict, reaction to natural causes such as pregnancy, and communicable diseases, in particular, HIV.

*Akuwam was out grazing goats, but she lost them; this led to a quarrel at home with her father, and it has given her some negative thoughts. Raia boy 2 (13).*

*Akuwam might have given her Kenya Certificate of Primary Education (KCPE) examination; unfortunately, she failed her exams, and now she is demoralized by the poor and unexpected results she got, which has contributed to her lack of appetite. School-going boy 5 (17).*

*I can say Akuwam felt hungry and starved, or their animals were raided by bandits, and that is what is causing her to have negative thoughts that result in her weakness, her not being interested in performing household chores, her sleepiness, and her being dull and lazy. Raia girl 1 (16).*

#### Schizophrenia

The second vignette involved Lokobe who was hearing voices and was convinced people were spying on him. He had kept to himself as a result. When asked what could be wrong with Lokobe, the adolescents mentioned curses, family conflict, effects of doing wrong, use of intoxicants and fear of exclusion from the society due to a communicable illness such as HIV. Incidentally, all the 'Raia' girls focused on guilt feelings as causing the symptoms Lokobe exhibited while not giving a name to the illness except that it was mental disturbance.

*According to me, Lokobe might have used bhang, and that made him hear unpleasant sounds and led to his hallucinations. Raia boy 4 (15).*

*Lokobe might have been affected by drugs, so he has been using drugs that affect his thinking. School-going boy 7 (17).*

*According to me, maybe Lokobe might have killed/raped someone, which made him guilty and made him feel like he was being followed by people even when he was alone. So he knows what he did was wrong. Raia girl 4 (16).*

#### Anxiety conceptualization

The last vignette involved Eragae who is always bothered by things more than they bother other people, and she is always worried and anxious. When asked what the problem was with Eragae, the adolescents talked of curses, feelings of guilt for wrongs done and fear of ostracization because of disease.

*According to me, Eregae might have been cursed by the people older than her, maybe because she has done something wrong to them. This is causing her a lot of fear. Raia boy 2 (13).*

*I guess Eregae killed someone, and she thinks people are tracking her. So, she feels that she is being tracked and followed all the time.* School-going boy 8 (17).

*I can say Eregae was a witch, and maybe she was caught by someone doing his things, which is making her nervous and feel that everyone is going to report her or reveal her behavior.* School-going girl 3 (17).

### Causes of the three conditions

#### Depression

When asked if they had ever felt like Akuwam and what caused such feelings, almost all participants said they had felt this sadness at certain times in their lives. They gave reasons such as sickness, hunger, family conflict, including being forced to leave school for marriage (school-going), evil spells cast by the jealous, sexual assault and family rejection.

*What causes such a situation like this one of Akuwam is when someone is hungry, that is, when they have no food to eat. This results in laziness, making someone lose energy. Sometimes when you sleep 3 days without eating food, that makes you become dull and not able to do any work or chores. And even sometimes when you are sick that is what causes such a situation. Raia girl 3(17).*

*Maybe someone insulted Akuwam saying she is HIV positive, which is what makes herisolate herself from other people, and it has really affected her normal behavior.* School-going girl 3 (17).

#### Schizophrenia causes

When asked what could have caused the condition, most adolescents said guilty feelings, and they noted that wrongdoing affects a person's emotions negatively.

*What causes it is when you have killed someone and that sticks in your mind and troubles you a lot. Raia boy 3 (13).*

*What might have caused such a situation is drug abuse, breaking of taboos and not performing rituals like attending the burial of the elderly, which leads to getting haunted by evil spirits.* School-going boy 5 (17).

*Lack of friends and people around you may cause that feeling simply because you may lack some advice from them. That makes you isolate yourself and have negative thoughts. Raia girl 5 (16).*

#### Anxiety causes

When asked what could have caused Eragae's situation and if they had ever felt that way, they had a variety of responses ranging from premonition to guilt feelings.

*What causes such a situation is when you are a witch and you are afraid of it being known. Raia boy 6 (14).*

*Stress has affected her thinking/mind. Maybe she was sexually assaulted thus affecting her mind leading to mental disturbance. She has been affected mentally.* School-going boy 5 (17).

*What causes Eregae's situation is the death of her family member. Maybe she lost either one or two of them. It can either be her mother or son, or something bad has happened to them, so that has made her feel nervous and look worried every time. Raia girl 1 (16).*

### Management of the three conditions

#### Depression management

When we asked the adolescents to suggest how similar situations to Akuwam's could be resolved, they mentioned help from family members and friends giving both emotional and physical help. They also noted that a person could use their own will power to pull themselves out of this situation.

*There was a time I lost a family member, I was depressed and stressed in that I couldn't even eat or sleep because he was my only source of hope, but my friend Esinyen advised me to leave such thoughts. Raia boy 4 (15).*

*One day I failed an exam. My teacher noticed and realized that I have never failed the way I did; he called me to his office and advised me to put more effort in the upcoming exam and that really helped me.* School-going boy 5 (17).

*If the issue is sexual harassment, the chief must be included to take an action and so should child rights experts, so they are the one to help them.* School-going boy 5 (17).

*When I was in that situation, I went to play and talk with my friends and I forgot what was stressing me. They advised me that that is normal, and I was able to feel relieved. Raia girl 1 (16).*

*There was a day when I was hungry, I was weak and my mother helped me. She went and brought me food.* School-going girl 4 (13).

#### Schizophrenia management

Asked how the schizophrenia condition can be managed, the participants mentioned friends, parents, neighbors and doctors as people most likely to help the affected person. The school-going participants added school counselors who were all teachers with the added responsibility of managing the counseling department.

*There was a time I stole someone's goat. I was nervous all the time and I was fearing that the owner of the goat could come for me. A friend of mine advised me to pay for someone's goat so that I become relieved. Raia boy 2 (13).*

*Such situations can be prevented by not being idle, and just by interacting with other people so that you avoid such negative thoughts that might affect you mentally and physically, simply because idleness is the devil's workshop.* School-going boy 1 (17).

*Such situations can be prevented by being around your friends so that you forget that something wrong might have happened to you. Raia girl 2 (17).*

#### Anxiety management

When asked how anxiety could be resolved, they mentioned prayers, friends, family, counselors, elders and teachers.

*When my relative was hospitalized, I thought he would die. I prayed for her and God helped him. Raia boy 2 (13).*

*Educating/Teaching people to stop stealing and having unnecessary sex. Raia boy 4 (15).*

*I took my friend who had a mental problem, with symptoms like fear. I took him for guidance and counseling in the hospital and he was helped.* School-going boy 2 (17).

*This can be prevented by giving self-advice. Such as asking yourself if I die, who will take care of my children? That gives you courage to ignore the situation and move on with your life regardless of the situation you were going through at that particular time. Raia girl 5 (16).*

*Such situations can be prevented by giving them advice on the importance of avoiding nervousness.* School-going girl 1 (16).

*Media talks like using FM and radios help them.* School-going girl 2 (13).

**Table 1.** Summary of study findings

|  | Depression (*Akiyalolong*) | Schizophrenia (*waarit/Ngikerep*) | Anxiety (*Ngatameta naaronok*) |
|---|---|---|---|
| Conceptualization | Extreme sadness<br>Mental illness, feelings of loss, feelings of being ostracized from the community, general illness | Curses, family conflict, effects of doing wrong, use of intoxicants, fear of exclusion from the society due to communicable illness such as HIV | Curse, disease, guilt |
| Causes | Sickness, hunger, family conflict, such as being forced to leave school for marriage (school-going), evil spells cast by the jealous, sexual assault and family rejection | Fear<br>Disease<br>Mental illness<br>Curses<br>Guilt feelings | Premonition, witchcraft, stress, grief, being a witch |
| Management | Counseling and advice from friends, teachers, family members and tangible help from the local chief, person's own will power through meditation | Friends, parents, neighbors, doctors, counselors (school) | Appeal for spiritual intervention, counseling from church leaders, meditation, media programs in FM stations |

## Discussions

### Key findings

We explored the mental health literacy of rural-based Kenyan adolescents regarding depression, schizophrenia and anxiety, and especially if they could recognize, know the causes and the management options available. In preparation to start field work, we had held discussions with some senior and well-respected members of the community, including the village elder, the chief and woman leader in the local area. These people, and later easily referred to by the adolescent participants, had given us local names for the three conditions we based our study on. They termed depression *Akiyalolong*', schizophrenia *Waarit/ Ngikerep* and anxiety *Ngatameta naaronok*. These were generally descriptive terms for the symptoms exhibited by the people suffering from the conditions (Table 1).

We read aloud vignettes of three people (Akuwam-girl, Lokobe-man and Eragae-girl) showing symptoms of depression, schizophrenia and anxiety disorder, respectively, and asked the adolescents what could be the problem, its causes and the management options for the symptoms. As we have discussed in our background, the proliferation of mental health illness, most of the time exhibited in incidents of harm both to self and to others, can only be adequately addressed if people are able to understand the illness, its causes and treatment.

### Conceptualization of depression, schizophrenia and anxiety

When asked what was wrong with Akuwam, Lokobe and Eragae, most adolescents gave their description of the symptoms exhibited by the three characters in the vignettes: Fear of exclusion from the community particularly due to deviant behavior, intoxicant use (especially in the case of schizophrenia), family conflict, curses extreme sadness and feelings of loss. There was no marked difference in how they conceptualized the three conditions.

One of the most cited descriptions of the three conditions in the vignettes involved deviant sexual relations: almost always rape, having sex outside a marital union, worrying that their sexual conduct has resulted in them contracting HIV and other sexually transmitted diseases and sexual rejection by a husband or partner. From these responses, we realized that sexual relations were very important for young Turkana people. This could be because as pastoralists (Krätli and Swift, 2014; Agol et al., 2020) the Turkana derive their livelihood from rearing domestic animals and consider owning animals an important part of who they are. This lifestyle means that they succeed when they have more children to take care of the animals (boys) and to bring in more animals through dowry (girls). Animal raiding and protection of domestic stock also dictate that a man has a large family. Procreation is therefore very important, and a family that cannot get more and more children is doomed as they will be poor and eventually be ostracized from the community and die off; the main reason why pastoralists are mostly polygamist (KNBS, 2014) and retain some children at home while a few are allowed to go to school. The retained girls can marry and both fetch dowry and start having more children, while the boys raise their animals and increase the stock through animal raids. In view of the preceding explanation, procreation and large families are the Turkana's everyday dreams and make it into the adolescents' discussions as some important things that can either build or destroy a person or community.

### Causes of depression, schizophrenia and anxiety

Participants thought that being excluded from the community for a variety of reasons, some of which are caused by deviant sexual relations, antisocial behavior such as stealing or having sexually communicable diseases, caused the three mental health conditions. The Turkana being a community that ascribe to a collectivist culture (Maquet, 1972; Richmond and Gestrin, 1998; Hofstede, 2001), in which they consider themselves as belonging to a whole where they are mandated to care for each other, ostracization from the community would have drastic negative effects on a person.

A few participants cited illness, both mental and physical, as the genesis of the symptoms exhibited in the vignettes. They did not mention any mental illness by name apart from referring to it as stress – a common colloquial Kenyan way of describing an overload of thoughts. The arid areas, such as the site of this study in northern Kenya Turkana County, have high levels of life stressors, including frequent animal raids from hostile neighbors from southern Sudan, in addition to livestock death due to incessant droughts. A person could be rich in animals one day and totally pauperized soon after because of such effects (Krätli and Swift, 2014). Lack of food resulting from poverty causes bodily weakness.

### Management of depression, schizophrenia and anxiety

When asked the possible management options for the three illnesses, the adolescents mostly mentioned counseling or, as they put it, 'talking to…' friends, followed by parents, then teachers and

conventional medical personnel could help with advice/counsel/guidance. By conventional, we hasten to add that we refer to both traditional and modern, read, Western ways of managing the conditions. Traditional ways of managing such conditions would involve consulting 'Emuron' (Lamphear, 1988), generally described as diviners, seers, medicine men and women who are still consulted today side-by-side with Western medicine. That most adolescents did not seek hospital-based help for the conditions corroborates the APHRC (2022) report noting that the use of health care services for emotional problems was low. In their article about alternative ways to engage mental illness in resource-scarce settings Ndetei et al. (2023) suggest that lay workers such as community health volunteers, peer counselors in school and teacher counselors could help school-going children with mental health issues that they have the capacity to manage.

## Conclusions

We explored mental health literacy among selected adolescents in northern Kenya and especially Turkana adolescents' conceptualization, and thoughts on the causes and management of depression, schizophrenia and anxiety. Our key finding was that all the adolescents unanimously acknowledged that anyone exhibiting the symptoms described in the vignettes was unwell physically, emotionally or both. When asked what the problem was with the characters in the vignettes, both *Raia* and school-going adolescents described what the characters were feeling. Only a few *Raia* recognized that the characters had a mental illness. This is an important step for any intervention project targeting the mental health wellbeing of adolescents in this community. Interventions with school-based adolescents have been documented, for instance, Osborne et al. (2021) conducted a randomized study with adolescents drawn from schools in Nairobi and Kiambu counties of Kenya with elevated symptoms on standardized depression and anxiety measures. They found out that the effect of the low-cost Shamiri intervention lasted longer by 7 months than the effect on the study skill control group.

The participants liberally used the local names given to the conditions and described what caused them in detail. The most cited cause of the conditions was fear of exclusion as a result of guilt after wrongdoing. That participants feared being excluded from community so much as to make them have 'walking nightmares' underscored the importance of social relations and support for successful existence in such rural communities. Any mental health intervention program should build on the locally available and well-esteemed social relations infrastructure to develop support for people suffering from mental health conditions.

Lastly, the participants revealed that spiritual leaders, parents, friends, teachers and selected community members could help people suffering from mental illness. A most notable suggestion was that a person with a mental illness could meditate to get well. The last suggestion shows how much power the adolescents thought they had, maybe because of living in such harsh terrain and conditions of armed conflict. They also suggested that the media through radio talk shows could also help listeners overcome mental illness. All these are important suggestions that future mental health interventionists could consider when engaging adolescents living in environments with similar characteristics to rural Lokichogio.

**Open peer review.** To view the open peer review materials for this article, please visit http://doi.org/10.1017/gmh.2023.25.

**Data availability statement.** Data are available on request from the corresponding author.

**Acknowledgments.** We would particularly like to thank the local chief, village elder, parents and adolescents of Lotoom 1 and Lotoom 2 villages in Lopiding sub-location of Lokichogio, Turkana County, for either helping mobilize participants or participating in the study. We are also indebted to the entire research team, including John Epuu, Lawrence Lobenyol and Simon Kouriang, who collected the data and participated in its transcription and translation. Patrick Njoroge, Maxwell Fundi and Michael Mumbo helped with the logistics and data management activities. P.W. was the PI in this project. She can be reached at pamela.wadende@gmail.com. T.S. was a co-PI in this project. Other supporting team members were John Epuu, Lawrence Lobenyol and Simon Esibital, who were research assistants. Triza Alimlim, Patrick Njoroge, Michael Mumbo, Maxwell Fundi and John Ng'asike helped with the data instrument translation process.

**Author contribution.** Both P.W. and T.S. contributed equally to this manuscript.

**Financial support.** The authors declare no financial support.

**Competing interest.** The authors declare no competing interest exists.

**Ethics standard.** The study was conducted according to the guidelines of the Declaration of Helsinki, and approved by Baraton University of Eastern Africa, Kenya (research ethics committee protocol code UEAB/IERC/12/03/2022 on 14-03-2022). Informed consent was obtained from parents of adolescents and assent was obtained from the adolescents involved in the study. We trained our research assistants on the informed consent process, including letting participants know their rights to withdraw at any time they experience distress. Research assistants were to let participants know that their responses would be kept confidential and only accessed by the research team and used in an anonymous form. The investigating team would also monitor participants' health and refer any who exhibited distress for psychological support to the local health center.

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
