## [Reviewer Report]

Chief Editor, 

Cambridge Prisms: Global Mental Health. 

30-1-23

Subject: Submission of manuscript for publication in Cambridge Prisms: Global Mental Health 

Dear Sir/Madam, 

I am enclosing a manuscript titled “Mental Health literacy: Perspectives from Northern Kenya Turkana adolescents” for possible publication. This manuscript has not been published, accepted for publication or is under review elsewhere. It is an original research manuscript. 

The significant finding of the study reported in this manuscript include that that adolescents in the rural Arid Northern Kenya pastoralist Turkana community have different understandings of the mental health conditions of depression, schizophrenia and anxiety. This has an impact on their health seeking behaviors and overall well-being. These findings are unique as they point to important options that can be harnessed for intervention to improve population mental health. 

This manuscript is based on a study conducted in rural Lokichogio town in Northern Kenya. 

Regards, 

P.A.W 

Pamela Wadende

---

## [Reviewer Report]

This is potentially important work that explores mental health literacy among adolescents in a part of Kenya that has not had many such studies.

I have identified several flaws in assumptions being made about causes of mental illness and its impacts. I have further concerns about the potential of reinforcing stigma that is evident in some of the initial sections, especially the abstract and background.

I am attaching brief comments and suggestions to improve the paper and I am happy to continue reviewing when these have been attended, please refer to comments below:

Abstract:

Authors write “In Kenya, school going adolescents

and children frequently exhibit violent outbursts such as arson attacks,

GBV and riots”

- It is not clear why this inference to “violent outbursts, arson attacks, gender-based violence and riots’ are being used to suggest a relationship with mental illness. Are the authors not (unintentionally) reinforcing stigma? Please review this statement and provide considered rationale for its inclusion in your abstract and paper, otherwise delete.

Abbreviations in abstract such as FGD, GBV

- Abbreviations must only be included if they are frequently used in your paper. In any case, write them in full first, then abbreviate in subsequent paragraphs/sections of the paper. Also, refer to Author Guidelines.

Authors write “Not much is documented for their nonschool

going counterparts yet they make up about 1.8 million.”

- It is unclear what populations this refers to; 

-Total of non-school going youth in Turkana, or the whole of Kenya

- Please revise your abstract so that even on its own, it tells a story to the reader before they can proceed reading the rest of your paper.

Impact statement:

Authors write:

“We conducted an ethnographic study to explore rural based Kenyan

mental health literacy for Depression, Schizophrenia and anxiety.”

- Earlier in the Impact Statement, Authors identify depression, anxiety, conduct disorders, and attention deficit/hyperactivity disorder as the leading causes of disease burden globally, yet for their study, they chose to focus on depression, anxiety and schizophrenia. Why were these three chosen and other mental disorders excluded?

Background:

The author/s write:

“Despite this documentation, not much attempt is made to link this destructive behavior to mental ill health.”

- Too many generalisations are being made in this paragraph. Reading the introduction to this paper leaves me thinking that there is a high level of personal opinion and lack of depth in well researched background information/literature review child and adolescent mental health, and impacts of mental illness. Instead, Authors have included statements that can only serve to reinforce stigma and stereotype. Why is the Daily Nation newspaper (Lines 24-27) being relied upon in analysis of this background section?

Please review your background section regarding use of grey literature and assumptions about causes and impacts of mental illness. 

Introduction:

Authors write:

“Concurrently, out-of-school youth also display behavior that points to mental ill health with reports that he youth commit the largest proportion of crime in Kenya.”

- There are too many lay assumptions and analysis in the paper that need to be revised. The fact that authors draw parallels between crime and mental illness reflects negatively on their understanding of the topic they have researched, and are writing about.

Methods:

Ethics and Participant supports pre- and post study

- I have noted that this study had ethics clearance by Baraton University.

Can authors provide evidence of risk mitigation strategies for participants in the study (pre- and post-). Ideally a statement of risk should have been added with this paper

---

## [Reviewer Report]

The theoretical underpinning or framework of this study needs to be stated with the reasons for adopting the particular theory.

So far, the introductory literature seems to be basic, descriptive, and passive with little or no researcher’s critical voice. Therefore, it would be useful to be critical and analytical throughout your engagement with the literature (i.e., identifying previous studies’ strengths and weaknesses and commenting on their implications on the specific and broader arguments you’re making in your proposed study). This is essential for the identification of key knowledge gaps/lacunas in the reviewed literature. This would also serve as the basis for the study’s rationale, statement of the research problem, research questions, aims and objectives, significance of the study, etc. (See Lines 8 through 35).

Method

Line 78

1. The study’s design could be reworded to articulate the mixed methods nature of the proposed study.

2. It would be beneficial too to rationalize why the use of both quantitative and qualitative research methods is essential for the proposed study and why either of the methods alone is not sufficient to examine/explore the topic area.

3. I suggest you validate your survey instruments within the population of the study.

4. It would also be beneficial to add a section on how your FGD questions were developed (e.g. the process adopted to ensure rigor).

General comment:

• The study’s preliminary triangulation seems to be slightly loose (e.g., ensure a strong link/triangulation from the project title, research questions, and study design).

• A thorough proofreading using the appropriate academic writing tone would be beneficial too. (eg lines 102, 107, 126,136, 146 154, etc. The use of ‘we’ should be replaced with connecting or transiting statements with the right flow in the best manner possible.

Note: Lines 126-132 are needless, similarly lines 146-152.

---

## [Reviewer Report]

This paper seeks to address an important subject - mental health literacy in the youth in one of the poorest parts of Kenya, and in a community that has continued to practice cultural traditions up to the present. One of the main research questions was about possibilities of support system for mental health. Qualitative methods were used. 

The literature review quotes an old reference 2013 which was on a clinical population and therefore the high prevalence (27.9%) of suicidal ideation. A lot has happened in non-clinical populations since then e.g. Journal of affective disorders. 2022 Apr 1;302:74-82., which gives more recent data on suicidal ideation and more recent publications on the same. The paper could be improved by making reference to data published in different Journals on mental health in Kenyan youth. The first reference in the list of references is on a report whose findings have not been reported in a peer reviewed publication i.e. not sure of the scientific fidelity on the report. 

It is not clear how objectively the different diagnoses were pegged to well established diagnostic criteria. Was it on DSM-V or ICD 10 or any other diagnostic criteria and then proceeded to find out what symptoms as described by the participants fitted into these criteria? This should come out clearly so that there is no doubt about what they call depression, schizophrenia or anxiety. 

Assuming these diagnoses were pegged to clearly defined diagnostic criteria, the contextualization and causes i.e. culturally appropriate explanatory models by the ethnic group are important, relevant and valid. But this is so only if the diagnoses were aligned to a set of diagnostic criteria. They are important because they would provide fertile grounds for sensitive and respectful engagement with the communities and the students on dialogue on other models and in particular the biopsychosocial model for explaining signs and symptoms and suggested intervention. Without this kind of dialogue, the communities and students will just continue with their own explanatory models. This will not be of much significance for intervention approaches. The paper would be greatly improved by this approach to understanding the conditions and management/interventions.

Finally, there is a lot documented literature on current thinking on how to provide cost-effective interventions within the school system in Kenyan context. For example, JAMA Psychiatry. 2021;78(8):829-837.

---

## [Reviewer Report]

Dear Editor,

We thank you for giving us this opportunity to resubmit our manuscript. We have responded to the reviewer’s comments both on the manuscript, highlighted in yellow, and as a letter to the reviewers which we have attached. Our word count has gone to 5599 but its is because of the line numbers and additional heading level instructions. 

We are willing to make any other corrections required of us.

Regards,

Pamela Wadende & Tholene Sodi

---

## [Reviewer Report]

Good work. Corrections effected. You may please check through for grammatical errors. All the best.

---

## [Reviewer Report]

Thank you for acting on reviewer suggestions, and/or providing rationales for structure and content of your paper. I am satisfied that all issues raised have been addressed and I recommend this paper for immediate publication.

Regards,

Elijah